# Impact of the DREAMS interventions on educational attainment among adolescent girls and young women: Causal analysis of a prospective cohort in urban Kenya

Sarah Mulwa[1,2]*, Lucy Chimoyi[3], Schadrac Agbla[4], Jane Osindo[2], Elvis O. Wambiya[2], Annabelle Gourlay[1], Isolde Birdthistle[1], Abdhalah Ziraba[2‡], Sian Floyd[1‡]

1 Faculty of Epidemiology and Population Health, London School of Hygiene & Tropical Medicine, London, United Kingdom, 2 African Population and Health Research Center, Nairobi, Kenya, 3 Research Management Department, The Aurum Institute, Johannesburg, South Africa, 4 Department of Health Data Science, University of Liverpool, Liverpool, United Kingdom

‡ AZ and SF are joint senior authors on this work.
* sarah.mulwa@lshtm.ac.uk

**Data Availability Statement:** Relevant data are within the manuscript and its Supporting Information files.

## Abstract

### Background

DREAMS promotes a comprehensive HIV prevention approach to reduce HIV incidence among adolescent girls and young women (AGYW). One pathway that DREAMS seeks to impact is to support AGYW to stay in school and achieve secondary education. We assessed the impact of DREAMS on educational outcomes among AGYW in Nairobi, Kenya.

### Methods and findings

In two informal settlements in Nairobi, 1081 AGYW aged 15−22 years were randomly selected in 2017 and followed-up to 2019. AGYW reporting invitation to participate in DREAMS during 2017–18 were classified as "DREAMS beneficiaries". Our main outcome was being in school and/or completed lower secondary school in 2019. We used multivariable logistic regression to quantify the association between being a DREAMS beneficiary and the outcome; and a causal inference framework to estimate proportions achieving the outcome if all, versus no, AGYW were DREAMS beneficiaries, adjusting for the propensity to be a DREAMS beneficiary. Of AGYW enrolled in 2017, 79% (852/1081) were followed-up to 2019. In unadjusted analysis, DREAMS beneficiaries had higher attainment than non-beneficiaries (85% vs 75% in school or completed lower secondary school, Odds Ratio (OR) = 1.9; 95%CI: 1.3,2.8). The effect weakened with adjustment for age and other confounders, (adjusted OR = 1.4; 95%CI: 0.9,2.4). From the causal analysis, evidence was weak for an impact of DREAMS (estimated 83% vs 79% in school or completed lower secondary school, if all vs no AGYW were beneficiaries, difference = 4%; 95%CI: -2,11%). Among AGYW out of school at baseline, the estimated differences were 21% (95%CI: -3,43%) among 15−17 year olds; and 4% (95%CI: -8,17%) among 18−22 year olds.

**Funding:** The evaluation of DREAMS is funded by the Bill and Melinda Gates Foundation (OPP1136774, http://www.gatesfoundation.org). Foundation staff advised the study team, but did not substantively affect the study design, instruments, interpretation of data, or decision to publish.

**Competing interests:** The authors have declared that no competing interests exist.

## Conclusions

DREAMS had a modest impact on educational attainment among AGYW in informal settlements in Kenya, by supporting both retention and re-enrolment in school. Larger impact might be achieved if more AGYW were reached with educational subsidies, alongside other DREAMS interventions.

## Introduction

Education, as a Sustainable Development Goal (SDG) is closely linked to other SDGs including good health and wellbeing (SDG 3), gender equality (SDG 5), and decent work and economic growth (SDG 8) [1–3]. The health, social, and economic benefits of educating adolescent girls and young women (AGYW) have been widely documented. Evidence suggests that more education among girls delays first sex, pregnancies, marriage and has cross-cutting benefits for maternal and child health [4–7]. Further, educating AGYW has been shown to reduce risk of HIV via modification of sexual behaviour, in addition to social and psychological changes like self-efficacy and empowerment [3, 8–10]. Spending more time in school might increase contact with health-promotion messages delivered within schools [8], and among girls limits opportunities to interact with male partners who are often older, and with a higher HIV risk profile compared to the girls [9, 11–13].

Even with these well-documented benefits of education, challenges that hinder access to primary education and transition to secondary school in sub-Saharan Africa (SSA) still exist. These constraints act at various levels such as within the families (e.g., inability to pay school fees, or pay for uniforms and supplies, or limited support from guardians), limited resources within schools and inequitable social norms at the community level where girls' education may be viewed as less important compared to boys' education [3, 14–16]. In most cases, those from low economic status and urban informal settlements are the most affected.

To address some of these constraints, several interventions have been implemented by national governments, non-governmental organisations and other funders [17–21]. Universal access policies through abolition of school fees for primary education have led to improvements in primary school enrolments in countries like Kenya, Tanzania and Uganda [22, 23]. Other interventions to support schooling in SSA take the form of cash transfers (conditional or unconditional) to school-going children and their families, or school support programs. A good proportion of studies evaluating these interventions–including a few studies on unconditional cash transfers–are conducted as randomized trials, and often assess the impact of a single component [10, 17, 20, 24–27]. In Zimbabwe for instance, a randomized study providing comprehensive school support (in form of fees, uniforms, and a school-based helper to monitor attendance and resolve problems) among orphans found that program beneficiaries were more likely to stay in school compared to those who did not. However, among those in school, there was no difference in academic performance between the beneficiaries and non-beneficiaries [25]. Very few evaluations have assessed the impact of comprehensive multi-component interventions on schooling outcomes [18, 21] in non-trial conditions.

Multi-component packages of interventions which simultaneously address multiple causes of adolescent vulnerabilities are increasingly promoted and delivered at scale through multi-sectoral collaborations, but little is known about their impacts in real world settings [28, 29]. With the population in SSA projected to triple in the next 80 years, there is a need to expand existing resources and infrastructure to ensure the health, educational, employment, and social

needs of the youthful population are met [30, 31]. The DREAMS (Determined, Resilient, Empowered, AIDS-free, Mentored and Safe lives) Partnership offers a package of evidence-based multi-sectoral interventions aimed at reducing HIV acquisition among AGYW. DREAMS is based on the principle that HIV prevention will be most effective when it targets the myriad of behavioural, social and structural factors driving HIV risk [19].

DREAMS recognises the inter-connectedness of educational and health outcomes, and strengthening the educational achievements of AGYW is an important part of the DREAMS theory of change. The DREAMS package includes interventions such as educational subsidies to support retention, promote return to school, and transition to secondary school [19]. Through curricula-based interventions in 'safe spaces', DREAMS also aims to enhance the agency of AGYW, to make and act upon strategic decisions to achieve their future goals, including educational goals. Access to resources and services such as modern methods of contraception, solar lamps, and supplies for menstrual management–through DREAMS–can help to reduce barriers to girls' education. The DREAMS package also includes interventions to strengthen families, for example, economically and through parenting support, and to mobilise communities more broadly to foster social norms that enhance gender equity and reduce gender-based violence [19]. These interventions are classified into primary interventions (considered a priority for AGYW in a given age group), secondary (based on individual 'need'), or contextual-level (**S1 Fig**). In an independent evaluation of DREAMS, we aimed to examine the combined effect of the DREAMS 'core package' on educational attainment among representative samples of AGYW in urban Kenya.

## Materials and methods

### Study context

The study was conducted in two urban informal settlements—Korogocho and Viwandani—in Nairobi, Kenya. Residents in these settlements experience numerous challenges including poor housing, inadequate access to clean water, high levels of food insecurity, poor infrastructure, and limited government services [32]. Adolescents living in these settings are at high risk of dropping out of school, especially in the transition period between primary and secondary school due to multiple reasons including high educational costs [14, 33]. Research also suggests that pregnancy is a reason for being out of school in these settings [34]. Government/public schools are few with an estimated 63% of primary school children attending non-government primary schools, which tend to be smaller, and less resourced in terms of teachers, services, facilities and amenities compared to public schools [35, 36]. Little information on secondary schools in informal settlements is available. However, data from the Ministry of Education shows that about 7 in 10 secondary schools in Nairobi County are private schools [37].

The current education system in Kenya consists of 8 years of primary school, 4 years of secondary school and 4 years of university education (a new curriculum is set to replace the current system). Learners sit for national exams at the end of primary and secondary cycles, with transitions to the next level (e.g., from primary to secondary) dependent on the performance in the primary-level examinations [15]. Students normally begin primary grade 1 at age six, and assuming they progress one grade each year, achieving full secondary education corresponds to 12 years of schooling, 10 years of schooling correspond to lower secondary education, and so on. While enrolment in the early grades of primary school is nearly universal, access to secondary school remains low in Kenya, with an estimated net enrolment ratio of 53% in 2018 [15].

## Study design

The DREAMS impact evaluation design and data collection protocol has been described in detail elsewhere [38]. In brief, we utilized three annual rounds of data collected from population-based, closed cohorts of randomly-selected samples of AGYW aged 10−14, 15−17 and 18 −22 years at the time of cohort recruitment (in 2017) in Korogocho and Viwandani informal settlements of Nairobi. The two settings were selected for inclusion in the impact evaluation of DREAMS given the established Nairobi Urban and Health and Demographic Surveillance System (NUHDSS) research platform in the area [39], which would enable timely evaluation. In the two evaluation settings, DREAMS interventions were introduced from early 2016, with one implementing partner (IP) coordinating the delivery of all interventions in each settlement.

The selected IPs were organizations with experience offering HIV related services or programs and were well-known within their respective communities. Implementation of interventions was staggered and newer services with no pre-existing infrastructure, e.g., social asset building, took a longer time to introduce and scale up, as IPs needed time for training and adapting the interventions to the local context. Educational subsidy programmes were integrated with government services and took considerable time to align and avoid duplication of beneficiaries. By March 2016 all services apart from Pre-Exposure Prophylaxis were being provided in Nairobi [40]. Although DREAMS was not randomised, invitation to participate was not offered to everyone. Rather, the implementers targeted and extended invitation to participate in DREAMS to the most vulnerable AGYW e.g., by inviting those who were food insecure, of school-age and out of school, or those who had ever been pregnant. Vulnerable AGYW were identified through the 'Girl Roster' census method, supplemented by local experience of community-based organisations [40, 41]. Invitation to participate in DREAMS continued into 2018, and intervention delivery continued during 2019–20.

At enrolment, we targeted a minimum sample of 500 girls in each age group (i.e., 500 aged 10−14 years and 1000 aged 15−22 years). Sample size was calculated to ensure statistical power to compare DREAMS and non-DREAMS beneficiaries across multiple outcomes and a range of assumptions about DREAMS uptake and impact [38]. All AGYW aged 10−22 years, and resident in the two settings were eligible for inclusion in the study. A randomly generated list of 1017 and 2599 girls aged 10−14 and 15−22 years respectively, was compiled from the most recent NUHDSS survey, and attempts were made to reach all girls on the list.

Data were collected using electronic interviewer-administered tools, developed by the research teams. The tools included modules on adolescent health and behaviour, educational expectations, schooling status and grade completed. DREAMS-specific questions covered self-reported invitation to participate in DREAMS activities and usage of each DREAMS intervention. Some measures, including aspirations and expectations were adopted from existing validated instruments (e.g., [42, 43]). The tool for 10−14 has been published elsewhere [44]. See **S1 Text** for an extract of questions among 15−22 years. Data collection activities were conducted between March–July 2017 (baseline), July–December 2018 (midline) and May–August 2019 (endline). In each interview round, at least three attempts were made so as to reach as many eligible participants as possible. Data were collected by trained field interviewers in face-to-face interviews. Among girls aged 10−14 years, interviews were conducted in a secure, private location at the field research office, with compensation provided to cover transport costs and snacks. Interviews among 15−22 year olds were mostly conducted in the AGYW's household. Because we used different tools, data among girls aged 10−14 years are analysed and summarised separately throughout this paper.

## Measures

**Outcome variables.**    All measures used in the present analyses are from the participants' self-reported data. Three outcome 'classes' were assessed: (a) drop-outs and re-enrolments, (b) retention and educational attainment, and (c) aspirations and expectations.

To explore school *drop-outs and re-enrolments*, we summarised the educational status of all participants at endline in 2019, conditional on their status at baseline. Participants were asked at each interview round if they were in school or not in school and the highest grade they had completed. We then created one variable with five mutually exclusive categories: (a) continued non-enrolment (out of school since baseline); (b) re-enrolment (out of school at baseline and in school at endline); (c) drop-out (in school at baseline, out of school at endline, and not completed secondary education); (d) school completion (in school at baseline, out of school at endline and completed secondary education); and (e) continued enrolment (in school since baseline).

To assess levels of *retention and educational attainment*, we used current schooling status and the highest level of education completed at endline in 2019 to create four composite variables that described educational *attainment* at endline as follows;

- (i) in school and/or have completed primary education (*attainment 1*)

- (ii) in school and/or have completed any post-primary education (*attainment 2*)

- (iii) in school and/or have completed lower secondary education (*attainment 3*)

- (iv) in school and/or have completed secondary education (*attainment 4*)

These outcomes allowed us to identify where gaps in educational progress may arise.

We also assessed *aspirations (*measured using eight items; Cronbach's α = 0.65) and *expectations (*eleven items; Cronbach's α = 0.82) about achieving certain life goals among all participants aged 15–22 years. However, we only present items specific to schooling in this paper. Educational aspirations captured how important attaining certain levels of education were to the AGYW. Future expectations included questions about the AGYW's perceived chances of completing secondary school or university. Among 10–14 year-olds, we analysed *aspirations* only (one item), as we did not collect data on *expectations* in this age group.

## Participation in DREAMS, and confounding variables

We used self-reported invitation to participate in DREAMS by 2018 (*yes or no*) to define a DREAMS beneficiary. We constructed directed acyclic graphs (causal diagrams) using Dagitty software [45] to represent the underlying causal structure of the relationship between being a DREAMS beneficiary, educational attainment, and other individual and household-level characteristics informed by our understanding of the context and how DREAMS targeting and implementation were done (**S2 Fig**). Age, study site (Viwandani or Korogocho), socio-economic status, food insecurity, marital status, pregnancy and sexual history, orphanhood status, and schooling status at the time of cohort enrolment were identified as potential confounders in this analysis. We also adjusted for highest education level of the household head and length of stay in the demographic surveillance area, as they were hypothesized to be predictive of the outcome.

## Statistical analysis

We summarised the proportions of AGYW who reported each outcome measure by age group and invitation to participate in DREAMS. We also compared DREAMS beneficiaries and non-beneficiaries with respect to important demographic characteristics.

The primary outcome for the causal analysis (described below) was being in school and/or completed lower secondary education in 2019 (*attainment 3*). Assuming the respondents begun primary grade 1 at age six [15], and that they progressed one grade each year, the majority of study participants aged 15–22 years at the time of cohort enrolment should have achieved *attainment 3* by endline in 2019. Almost all girls aged 10–14 years at the time of cohort enrolment were in school at endline, therefore analyses of DREAMS' impact were only conducted among AGYW aged 15–22 years. Our causal question of interest was whether DREAMS improved educational *attainment 3* among AGYW aged 15–22 years at cohort enrolment. We used a staged approach to answer this question. In the first step, we assessed the associations between being a DREAMS beneficiary and *attainment 3* using a multivariable logistic regression model, adjusting for the confounding variables identified in the causal diagram. From this model, we present the unadjusted, age adjusted, and fully adjusted Odds Ratios (OR), with their respective 95% Confidence Intervals (CIs).

We then conducted analysis within a causal inference framework to compare the percentage of AGYW in school and/or completed lower secondary education in 2019 (*attainment 3*), under two counterfactual scenarios that all AGYW were invited to DREAMS versus none were invited to DREAMS (the causal assumptions are summarised in **S2 Text**). Our primary analysis approach was propensity-score regression adjustment. We chose to use propensity scores because of flexibility, and the fact that the approach reduces the number of explanatory variables (and therefore the number of regression parameters) estimated from the final model [46, 47]. This was operationalised as follows. First, the outcome of the propensity score model was invitation to DREAMS by 2018 (yes or no), with explanatory variables identified from the causal diagram as confounding variables and also including those for which there was evidence they were independent predictors of educational attainment. This model was used to estimate a "propensity to be invited to DREAMS" for each AGYW (**S3 Fig**).

We then fitted a logistic regression model to predict the probability of *attainment 3*, first with restriction to AGYW who were DREAMS beneficiaries, adjusting for the estimated propensity score and age group. From this model, we predicted the probability of the outcome for *all* AGYW, irrespective of whether or not they were a DREAMS beneficiary. The average value of these probabilities was used to estimate the percentage of AGYW with *attainment 3* under the counterfactual scenario that all AGYW were DREAMS beneficiaries. We repeated this approach for AGYW who were not DREAMS beneficiaries, to estimate the percentage of AGYW with the outcome under the counterfactual scenario that no AGYW were DREAMS beneficiaries. We present these average predictions overall and separately for younger and older AGYW.

Our primary effect measure was the difference in the average predicted probability of achieving *attainment 3* between the two hypothetical scenarios above. Confidence intervals were generated using a bootstrap procedure, repeating the estimation procedure described above in 1000 samples taken with replacement from the complete dataset.

As the impact of DREAMS may have varied by whether or not an AGYW was in school at the start of the intervention, we conducted a pre-specified sub-group analysis for *attainment 3* separately for AGYW aged 15–22 years who were in school, as well as among those who were out of school at baseline, following the same approach as described above. We further conducted a post-hoc analysis among AGYW who had not completed lower secondary education at baseline. All analyses were restricted to study participants followed up at endline, and were conducted in Stata/SE 15.1 software (StataCorp, College Station, TX).

## Sensitivity analyses

To examine the robustness of our estimate of the impact of DREAMS on *educational attainment* obtained from propensity score regression adjustment, we conducted alternative analyses

for comparison, namely: propensity score stratification, "inverse probability of treatment weighting" (with the probability of treatment being estimated by the propensity score), multi-variable outcome regression adjustment, and per protocol analysis (based on invitation to DREAMS and also the number of primary interventions accessed). For all these analyses, we present our primary effect measure (difference), with the respective 95% CIs. We assessed covariate balance statistics using the inverse-probability-of-treatment weighting approach, and findings indicate sufficient balance was achieved after the weighting (**S2 Text**).

### Ethics approval

Ethics approval was obtained from African Medical and Research Foundation (AMREF) Health Africa Ethics and Scientific Review Committee (ESRC) (AMREF; No ESRC P298/2016) and the London School of Hygiene & Tropical Medicine (LSHTM; Ref 11835). An information sheet was used to provide and discuss details about the study with potential participants and their parents/guardians, before requesting written consent to participate. For participants under age 18, written informed parental/guardian consent and participant assent were obtained before commencing an interview.

## Results

### Demographic characteristics of the AGYW

A total of 606 girls aged 10–14 years (response rate of 89%, n = 684 eligible) and 1081 aged 15–22 years (response rate of 61%, n = 1770 eligible) were enrolled into the cohort study at baseline, among those randomly selected from the database. Retention rates were high, with 82% among those aged 10–14 years (494/606) and 79% among those aged 15–22 years (852/1081) followed up at endline. Among AGYW aged 15–22 years, retention was higher among those who had been invited to participate in DREAMS at the time of cohort enrolment, with a larger difference among older (aged 18–22 years) than younger AGYW (aged 15–17 years). Older AGYW, those from Viwandani, and those out of school were less likely to be followed-up (**S1 Table**).

Of the 494 girls aged 10–14 years at enrolment, almost all were attending school (99%) and a higher proportion were aged 10–12 years (62%) than 13–14 years at the time of enrolment into the study. Seventy-seven percent reported that they had been invited to participate in DREAMS interventions by 2018. Participant baseline characteristics were broadly similar by invitation status except for the study site (**Table 1**).

Among AGYW aged 15–22 years at enrolment and followed-up (n = 852), the majority were aged 15–17 years (55%), were in school (63%,) and were residents of the demographic surveillance area since birth (52%) at the time of enrolment into the study. Primary school completion levels were high at baseline, with 90% having completed primary education. Seventy-four percent of the participants had been invited to participate in DREAMS interventions by 2018. DREAMS beneficiaries were more likely to be younger, in school and food insecure at enrolment compared to non-DREAMS beneficiaries (**Table 2**). These patterns, among those who were followed up, are similar to those described at baseline among all who were enrolled to the cohort [48].

### Uptake of DREAMS interventions

The majority of DREAMS beneficiaries (≥80% across various sub-groups) received multiple primary interventions by 2019. For example, among 15–17 year olds, 86% of the DREAMS beneficiaries accessed ≥3 (out of 7) primary interventions (**S2 Table**). The proportion of

**Table 1. Enrolment profile (characteristics at cohort enrolment) among girls aged 10–14 years and followed up in 2019, by invitation to participate in DREAMS.**

| Characteristics at enrolment | Overall | Never invited | Invited by 2018 | p-value |
|---|---|---|---|---|
| | N = 494 | N = 114 (23.1) | N = 380 (76.9) | |
| | n (%) | n (%) | n (%) | |
| **Age group (years)** | | | | |
| 10–12 | 307 (62.1) | 71 (62.3) | 236 (62.1) | |
| 13–14 | 187 (37.9) | 43 (37.7) | 144 (37.9) | 0.973 |
| **Informal settlement area** | | | | |
| Korogocho | 280 (56.7) | 52 (45.6) | 228 (60.0) | |
| Viwandani | 214 (43.3) | 62 (54.4) | 152 (40.0) | 0.007 |
| **Currently enrolled in school** | | | | |
| No | 4 (0.8) | 3 (2.6) | 1 (0.3) | |
| Yes | 490 (99.2) | 111 (97.4) | 379 (99.7) | 0.040 |
| **School progress** | | | | |
| 2+ classes behind | 150 (30.4) | 31 (27.2) | 119 (31.3) | |
| <2 classes behind | 344 (69.6) | 83 (72.8) | 261 (68.7) | 0.401 |
| **Orphanhood status** | | | | |
| Not an orphan | 428 (86.6) | 100 (87.7) | 328 (86.3) | |
| Single/double orphan | 66 (13.4) | 14 (12.3) | 52 (13.7) | 0.699 |
| **Paid jobs/activities, last 6 months** | | | | |
| No | 470 (95.1) | 106 (93.0) | 364 (95.8) | |
| Yes | 24 (4.9) | 8 (7.0) | 16 (4.2) | 0.221 |
| **Family food insecurity[a]** | | | | |
| Never | 188 (38.1) | 53 (46.5) | 135 (35.5) | |
| Sometimes | 267 (54) | 55 (48.2) | 212 (55.8) | 0.089 |
| Often | 39 (7.9) | 6 (5.3) | 33 (8.7) | |
| **Romantic relationships** | | | | |
| Never been in a relationship | 445 (90.3) | 100 (87.7) | 345 (91.0) | |
| Ever been in a relationship | 48 (9.7) | 14 (12.3) | 34 (9.0) | 0.498 |
| **Sexually exploited[b]** | | | | |
| No | 463 (93.7) | 107 (93.9) | 356 (93.7) | |
| Yes | 31 (6.3) | 7 (6.1) | 24 (6.3) | 0.946 |
| **Physical violence, last 6 months** | | | | |
| No | 414 (83.8) | 93 (81.6) | 321 (84.5) | |
| Yes (being slapped, hit, physically hurt) | 80 (16.2) | 21 (18.4) | 59 (15.5) | 0.462 |
| **Verbal violence, last 6 months** | | | | |
| No | 327 (66.2) | 72 (63.2) | 255 (67.1) | |
| Yes (teased, bullied or threatened) | 167 (33.8) | 42 (36.8) | 125 (32.9) | 0.435 |

[a]ever been a time when your family did not have enough food because they had no money.

[b]reported being threatened, coerced or being forced into being touched or having (first) sex, or said they were unwilling to have (first) sex, or they were ever forced into/attempted sex by an adult (childhood experiences), or reported being touched in the last 6 months in a way they did not want to be touched.

DREAMS beneficiaries accessing educational subsidies were 57%, 53% and 20% among AGYW aged 10–14 years, 15–17 years and 18–22 years, respectively. Among AGYW aged 15–22 years out of school at baseline, only 4% of the DREAMS beneficiaries accessed educational subsidies by 2019. Among those in school at baseline, proportions accessing educational subsidies were significantly higher among DREAMS beneficiaries (56%; 239/425), compared to non-beneficiaries (20%; 23/115). Uptake of each DREAMS core-package intervention is summarised in detail elsewhere [49].

**Table 2. Enrolment profile (characteristics at cohort enrolment) among AGYW aged 15–22 years and followed up in 2019, by invitation to participate in DREAMS.**

| Characteristics at enrolment | Overall | Never invited | Invited by 2018 | p-value |
|---|---|---|---|---|
| | N = 852 | N = 224 (26.3) | N = 628 (73.7) | |
| | n (%) | n (%) | n (%) | |
| **Age, pregnancy and marital status** | | | | |
| 15–17 years | 464 (54.5) | 95 (42.4) | 369 (58.8) | |
| 18–22:never married, never pregnant | 201 (23.6) | 59 (26.3) | 142 (22.6) | |
| 18–22:never married, ever pregnant | 40 (4.7) | 10 (4.5) | 30 (4.8) | <0.001 |
| 18–19:ever married and ever pregnant | 32 (3.8) | 14 (6.3) | 18 (2.9) | |
| 20–22:ever married and ever pregnant | 115 (13.5) | 46 (20.5) | 69 (11) | |
| **DSS study site** | | | | |
| Korogocho | 513 (60.2) | 143 (63.8) | 370 (58.9) | |
| Viwandani | 339 (39.8) | 81 (36.2) | 258 (41.1) | 0.196 |
| **Highest level completed** | | | | |
| Less than primary grade 8 | 89 (10.4) | 29 (12.9) | 60 (9.6) | |
| Primary grade 8 or more | 763 (89.6) | 195 (87.1) | 568 (90.4) | 0.154 |
| **Orphanhood status** | | | | |
| Not an orphan | 663 (77.8) | 170 (75.9) | 493 (78.5) | |
| Single/double orphan | 189 (22.2) | 54 (24.1) | 135 (21.5) | 0.419 |
| **Food insecure** | | | | |
| No | 564 (66.2) | 166 (74.1) | 398 (63.4) | |
| Yes | 288 (33.8) | 58 (25.9) | 230 (36.6) | 0.004 |
| **Self-assessed household poverty** | | | | |
| Very poor | 115 (13.5) | 23 (10.3) | 92 (14.6) | |
| Moderately poor | 672 (78.9) | 180 (80.4) | 492 (78.3) | 0.161 |
| Not poor | 65 (7.6) | 21 (9.4) | 44 (7) | |
| **Wealth quantile** | | | | |
| Poor | 303 (35.6) | 77 (34.4) | 226 (36) | |
| Medium | 277 (32.5) | 79 (35.3) | 198 (31.5) | 0.587 |
| Wealthy | 272 (31.9) | 68 (30.4) | 204 (32.5) | |
| **How long have you stayed in the DSA[a,b]** | | | | |
| Since birth | 446 (52.3) | 93 (41.5) | 353 (56.2) | |
| 0–5 years | 173 (20.3) | 73 (32.6) | 100 (15.9) | |
| 6–10 years | 110 (12.9) | 31 (13.8) | 79 (12.6) | <0.001 |
| 10+ years | 123 (14.4) | 27 (12.1) | 96 (15.3) | |
| **Highest educational level of the household head[b]** | | | | |
| None/incomplete primary | 202 (23.7) | 41 (18.3) | 161 (25.6) | |
| Incomplete secondary | 285 (33.5) | 67 (29.9) | 218 (34.7) | 0.011 |
| Complete secondary/tertiary | 252 (29.6) | 77 (34.4) | 175 (27.9) | |
| Don't know | 113 (13.3) | 39 (17.4) | 74 (11.8) | |
| **Currently enrolled in school** | | | | |
| No | 312 (36.6) | 109 (48.7) | 203 (32.3) | |
| Yes | 540 (63.4) | 115 (51.3) | 425 (67.7) | <0.001 |

[a]DSA—Demographic Surveillance Area

[b]these variables were not a targeting criteria for DREAMS, included because they are predictors of the outcome.

# Education outcomes among 10–14 year-olds

**Aspirations.** Aspirations about schooling were high, with >90% stating that they thought they would complete university or college in each year of follow-up. Among DREAMS

beneficiaries, aspirations slightly increased over time, but the differences (compared to non-beneficiaries) were small (**S4 Fig; Panel A).**

**Retention and educational attainment.** Of the 494 girls followed up, 97% (n = 480) were still enrolled in school at endline. Proportions in school were slightly higher among DREAMS beneficiaries compared to non-beneficiaries (98% (372/380) vs 95% (108/114), respectively). At endline, 60% of all girls, and 93% (164/177) among those aged 13–14 years were enrolled in primary grade 8 or secondary school.

Of the 14 who were out of school at endline, 14% had not completed primary education, 64% (9/14) had completed primary school but had not transitioned to a higher level, while 21% dropped out before completing secondary education. Lack of school fees (n = 9) and pregnancy (n = 4) were the two most cited reasons for not being in school.

## Education outcomes among 15–22 year-olds

**Aspirations and expectations.** Generally, *aspirations* regarding finishing secondary school and going to college/university among 15–22 year olds were high both at baseline and at endline. For instance, 80% of DREAMS beneficiaries and 76% of non-beneficiaries rated going to university 'very important' at endline. Aspirations tended to be higher among DREAMS beneficiaries compared to non-beneficiaries, although the differences were small (**S4 Fig; Panel B).**

*Expectations* about schooling were in general lower than the aspirations. *Expectations* about finishing secondary school were stable or increased slightly over time. Expectations regarding going to university were low, with <50% of the DREAMS beneficiaries in each age group saying that their chances of going to university were 'high' (**Fig 1**).

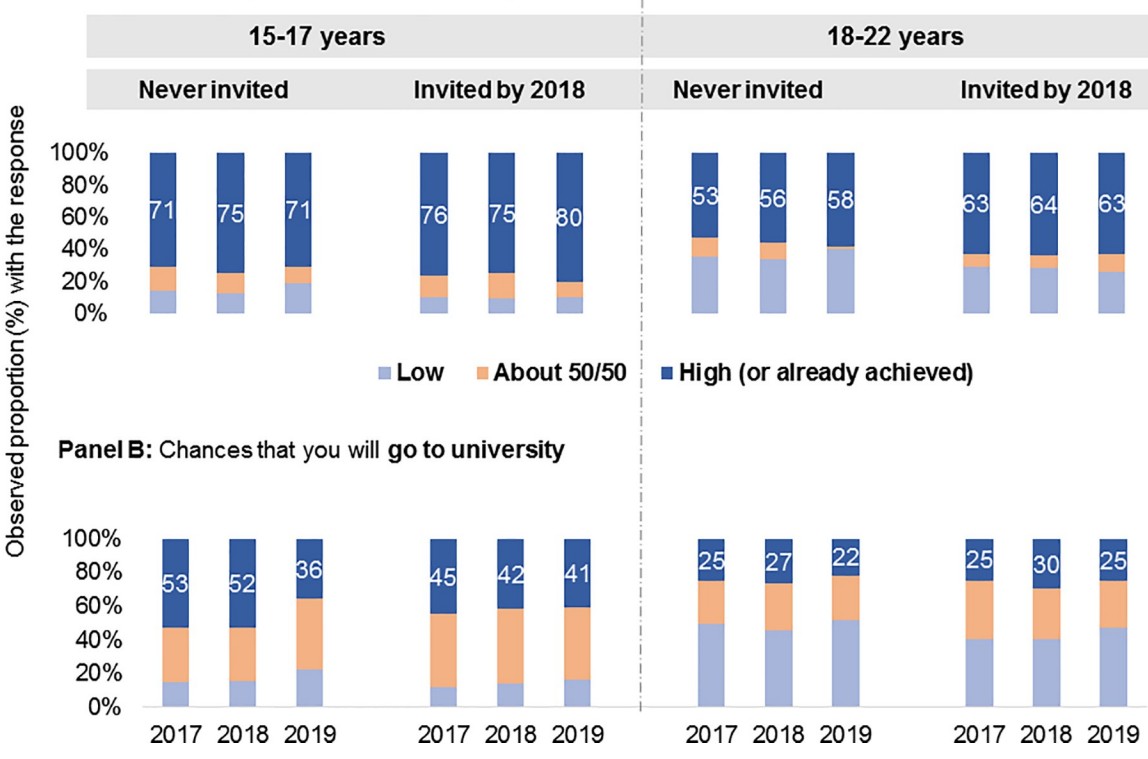

**Fig 1. Expectations about schooling among AGYW aged 15–22 years.**

**Table 3. Educational status at endline among AGYW aged 15–22 years at cohort enrolment in 2017, by invitation to DREAMS.**

| Schooling status at endline (in 2019) | All AGYW aged 15–22 years | | | 15–17 years | | 18–22 years | |
|---|---|---|---|---|---|---|---|
| | Overall | Never invited | Invited by 2018 | Never invited | Invited by 2018 | Never invited | Invited by 2018 |
| | N = 852 | N = 224 | N = 628 | N = 95 | N = 369 | N = 129 | N = 259 |
| | n (%) | n (%) | n (%) | n (%) | n (%) | n (%) | n (%) |
| Continued non-enrolment (out of school since baseline) | 254 (29.8) | 97 (43.3) | 157 (25.0) | 17 (17.9) | 31 (8.4) | 80 (62.0) | 126 (48.6) |
| Re-enrolment during follow-up | 40 (4.7) | 7 (3.1) | 33 (5.3) | 1 (1.1) | 12 (3.3) | 6 (4.7) | 21 (8.1) |
| Dropout during follow-up[a] | 42 (4.9) | 5 (2.2) | 37 (5.9) | 2 (2.1) | 27 (7.3) | 3 (2.3) | 10 (3.9) |
| Completed secondary education during follow-up | 175 (20.5) | 45 (20.1) | 130 (20.7) | 18 (18.9) | 72 (19.5) | 27 (20.9) | 58 (22.4) |
| Continued enrolment (in school since baseline) | 341 (40.0) | 70 (31.3) | 271 (43.2) | 57 (60.0) | 227 (61.5) | 13 (10.1) | 44 (17.0) |

[a]Out of the 42, 40% (n = 17) dropped out before completing lower secondary education; while (60%, n = 25) dropped out after completing lower secondary (but before completing secondary school).

**Retention and educational attainment.** The education status of the participants at endline, distinguishing between those who were in or out of school at baseline, is summarised in **Table 3**. Of the 852 followed up, 40% were in school at baseline and remained so throughout the follow-up period, ~30% were out of school at baseline and remained so, ~5% were re-enrolled during the follow-up, ~5% dropped out before completing secondary education, and ~20% completed secondary education during follow-up. Overall, DREAMS beneficiaries were more likely to remain in school throughout the follow-up period (43%) compared to non-DREAMS beneficiaries (31%).

At endline, proportions who had completed some post-primary education, and lower secondary education were generally higher among DREAMS beneficiaries compared to non-beneficiaries for both younger and older AGYW (**Fig 2**).

In the unadjusted logistic regression analysis, a higher percentage of DREAMS beneficiaries than non-beneficiaries reported being in school and/or completing lower secondary education, at 85% compared with 75% (crude Odds Ratio (cOR) = 1.9; 95%CI: 1.3–2.8). This effect weakened with adjustment for age and other confounders (adjusted OR (aOR) = 1.4; 95%CI: 0.9–2.4) (**Table 4A**).

Based on the propensity-score regression adjusted analysis, overall, we estimated that proportions in school and/or completing lower secondary education would increase from 79% if none were DREAMS beneficiaries to 83% if all were DREAMS beneficiaries (difference = 4%; 95%CI: -2 to 11%) (**Table 5A**). When analyses were stratified by age group, the magnitude of change was similar to the overall finding; among younger AGYW aged 15–17 years at enrolment an estimated difference of 5%; (95%CI: -2 to 14%) and among older AGYW (18–22 years) at enrolment an estimated difference of 3%; (95%CI: -7 to 13%]).

## Sub-group analysis according to schooling status at baseline

Sixty-five percent of those out of school at baseline (203/312) and 79% of those in school at baseline (425/540), were DREAMS beneficiaries. Baseline characteristics for these two sub-groups were broadly similar among those invited or not to DREAMS, although among AGYW in school at baseline, DREAMS beneficiaries were more likely to be food insecure compared to non-DREAMS beneficiaries (**S3 Table**).

Only 58% of the AGYW out of school at baseline had completed any post-primary training, with proportions significantly higher among AGYW aged 18–22 years (64%) compared to

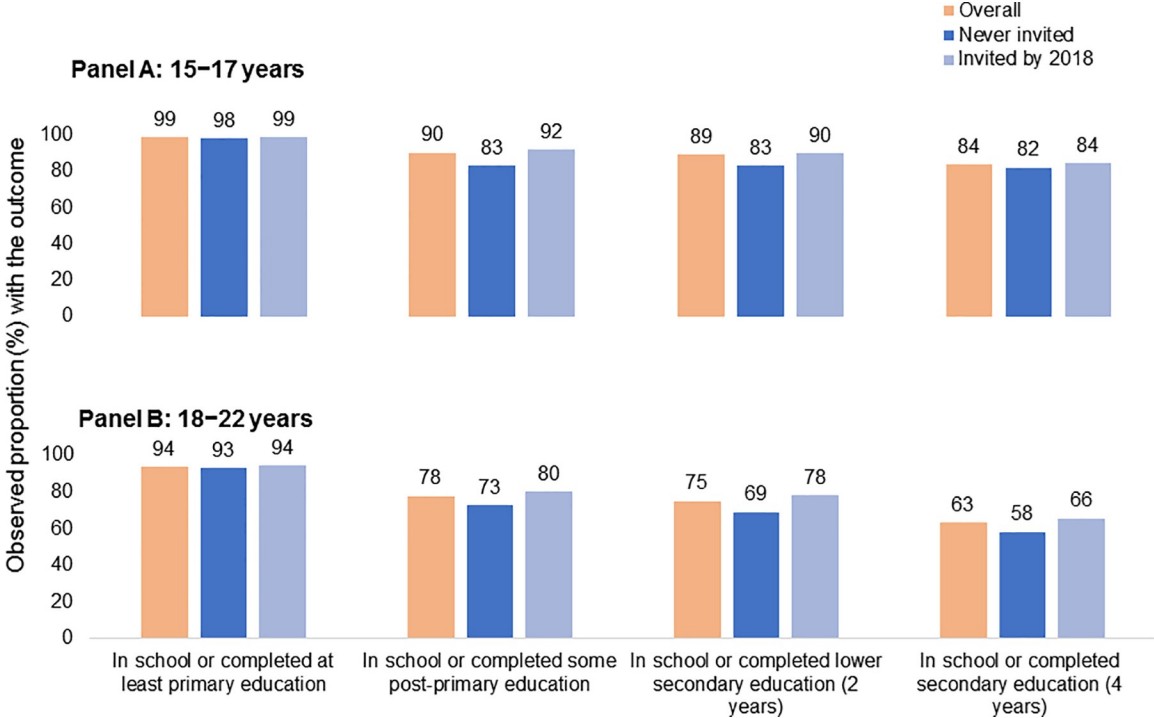

**Fig 2. Educational attainment at endline among AGYW aged 15–22 years by invitation to DREAMS and age at enrolment.** Denominators: Among 15–17 years, total N = 464; invited by 2018 N = 369; Among 18–22 years, total N = 388; invited by 2018 N = 259.

those aged 15–17 years (36%) (**S5 Fig**). Proportions in school at endline were significantly higher among DREAMS beneficiaries compared to non-beneficiaries (16% (33/203) vs 6% (7/109)).

From the multivariable logistic regression analysis among AGYW out of school at baseline, there was only weak evidence of an effect of DREAMS overall (aOR = 1.5; 95%CI: 0.9–2.5) or among AGYW aged 18–22 years (aOR = 1.4; 95%CI: 0.8–2.6). There was evidence of an effect of DREAMS among younger DREAMS beneficiaries (aged 15–17 years at baseline) with 45% vs 25% in school and/or having completed lower secondary school by 2019 and an adjusted OR of 4.6 (95%CI: 1.1–18.9) (**Table 4B**). In the propensity-score adjusted regression analysis, DREAMS was estimated to increase proportions in school and/or completing lower secondary education from 24% (95%CI: 8–46%) among 15–17 year olds if none were DREAMS beneficiaries to 45% (95%CI: 31–60%) if all were beneficiaries (difference = 21%; (95%CI: -3 to 43%)). The estimated effect was small among AGYW aged 18–22 years (difference = 4%; (95% CI: -8 to 17%)) (**Table 5B**). The findings from these analyses were consistent with those from sensitivity analyses (**S4 Table**). The vast majority (≥95%) of AGYW who were in school at baseline remained in school or had completed lower secondary education by endline in 2019, with little evidence of an impact of DREAMS (**Tables 4C and 5C**). Similarly, we did not find evidence for an impact of DREAMS when analyses were restricted to those who had not completed lower secondary education at baseline (**Tables 4D and 5D**).

## Discussion

This study provides insights into school enrolment, levels of attainment, and the impact of a complex intervention delivered at scale in representative samples of AGYW in urban informal settlements of Kenya. Results indicate that almost all young adolescents aged 10–14 years at

**Table 4. (a, b, c, d) Association between DREAMS and educational attainment** ** **among AGYW aged 15–22 years using multivariable logistic regression, overall and stratified by age at enrolment.**

| | Not a DREAMS beneficiary (N/%) | DREAMS beneficiary (N/%) | % non-beneficiaries in school or completed lower secondary education | % beneficiaries in school or completed lower secondary education | % Difference (un-adjusted) | Unadjusted Odds Ratio (95% CI[a]) | Age Adjusted Odds Ratio (95% CI) | Fully Adjusted Odds Ratio (95% CI) | p-value (LR-test)[b] |
|---|---|---|---|---|---|---|---|---|---|
| **a) Full sample[c]** | | | | | | | | | |
| Overall | 224 (26.3) | 628 (73.7) | 75.0 | 85.4 | 10.4 | 1.9 (1.3–2.8) | 1.7 (1.1–2.4) | 1.4 (0.9–2.4) | 0.173 |
| 15–17 Years | 95 (20.5) | 369 (79.5) | 83.2 | 90.5 | 7.3 | 1.9 (1.0–3.7) | 1.8 (0.9–3.4) | 1.6 (0.6–4.4) | 0.338 |
| 18–22 Years | 129 (33.3) | 259 (66.8) | 69.0 | 78.0 | 9.0 | 1.6 (1.0–2.6) | 1.5 (1.0–2.5) | 1.5 (0.7–2.8) | 0.286 |
| **b) Sub group analysis: among those out of school at baseline[d]** | | | | | | | | | |
| Overall | 109 (34.9) | 203 (65.1) | 51.4 | 60.6 | 9.2 | 1.5 (0.9–2.3) | 1.6 (1.0–2.6) | 1.5 (0.9–2.5) | 0.163 |
| 15–17 Years | 20 (29.8) | 47 (70.2) | 25.0 | 44.7 | 19.7 | 2.4 (0.8–7.8) | 2.9 (0.9–10.1) | 4.6 (1.1–18.9) | 0.032 |
| 18–22 Years | 89 (36.3) | 156 (63.7) | 57.3 | 65.4 | 8.1 | 1.4 (0.8–2.4) | 1.5 (0.9–2.5) | 1.4 (0.8–2.6) | 0.301 |
| **c) Sub group analysis: among those in school at baseline[d]** | | | | | | | | | |
| Overall | 115 (21.3) | 425 (78.7) | 97.4 | 97.2 | -0.2 | 0.9 (0.3–3.3) | 0.8 (0.2–3.1) | 1.1 (0.3–4.7) | 0.900 |
| 15–17 Years | 75 (18.9) | 322 (81.1) | 98.7 | 97.2 | -1.5 | 0.5 (0.1–3.8) | 0.5 (0.1–3.8) | 0.9 (0.1–7.5) | 0.887 |
| 18–22 Years | 40 (28.0) | 103 (72.0) | 95.0 | 97.1 | 2.1 | 1.8 (0.3–10.9) | 1.6 (0.2–10.2) | 1.6 (0.2–13.0) | 0.649 |
| **d) Sub group analysis: among those who had not completed lower secondary school at baseline (post-hoc)[d,e]** | | | | | | | | | |
| Overall | 108 (26.9) | 293 (73.1) | 49.1 | 68.6 | 19.5 | 2.3 (1.4–3.6) | 1.7 (1.0–3.1) | 1.2 (0.5–3.0) | 0.630 |
| 15–17 Years | 64 (22.3) | 223 (77.7) | 75.0 | 84.3 | 9.3 | 1.8 (0.9–3.5) | 1.8 (0.9–3.5) | 1.2 (0.4–3.7) | 0.707 |
| 18–22 Years | 44 (38.6) | 70 (61.4) | 11.4 | 18.6 | 7.2 | 1.9 (0.6–5.4) | 1.9 (0.6–5.4) | 1.7 (0.3–9.1) | 0.523 |

**Educational attainment is a binary variable taking values 1 if an individual was in school or had completed lower secondary school at endline (for those out of school) and 0 otherwise

[a]CI—Confidence Interval

[b]LRT—Likelihood Ratio Test

[c]Final model adjusted for age, pregnancy and marital history (composite), study site, highest grade completed at baseline, orphanhood status, self-assessed poverty, wealth quantile, food insecurity, length of stay in the demographic surveillance area, education level of the household head and schooling status at baseline

[d]sub-group analyses adjusted for fewer variables (reduced sample): age, pregnancy and marital history (composite), study site, wealth quantile, food insecurity, education level of the household head

[e]Final model further adjusted for schooling status at baseline.

baseline in 2017 were still in school at endline in 2019, and 60% had completed at least seven years of schooling. Among AGYW aged 15–22 years at baseline, virtually all had completed primary education. However, only about 60% of those out of school at baseline had accessed any post-primary training. From the causal analyses, our findings indicate an overall modest increase in completing at least two years of secondary education or currently being in school of 4% due to DREAMS. We found high levels of aspirations but lower expectations about schooling.

AGYW's aspirations about schooling were high, but expectations of what is realistically attainable were lower, consistent with findings from elsewhere. Studies in Spain, the United States, and Kenya have reported this mismatch in educational aspirations and expectations, and suggested that the mismatch is higher among people from low socio-economic groups than those from more socio-economically well-off groups [43, 50–53].

**Table 5. (a, b, c, d) Estimated causal effect of DREAMS on educational attainment, from regression analysis with adjustment for the 'propensity to be a DREAMS beneficiary'.**

| | % in school or completed lower secondary education in total study population | Estimated % in school or completed lower secondary education if none benefit from DREAMS: % (95% CI[a]) | Estimated % in school or completed lower secondary education if all benefit from DREAMS: % (95% CI) | % Difference (95% CI) | Odds Ratio (95% CI) |
|---|---|---|---|---|---|
| **a) Full sample[b]** | | | | | |
| Overall | 82.6 | 79.2 (72.9–84.3) | 83.4 (80.3–86.5) | 4.2 (-1.8 to 11.1) | 1.3 (0.9 −2.0) |
| 15–17 Years | 89.0 | 84.9 (76.6–91.2) | 89.9 (86.5–92.9) | 5.1 (-2.4 to 13.7) | 1.6 (0.8 −3.0) |
| 18–22 Years | 75.0 | 72.5 (64.3–80.4) | 75.6 (70.2–81.2) | 3.1 (-6.5 to 13.2) | 1.2 (0.7 −1.9) |
| **b) Sub group analysis: among those out of school at baseline[c]** | | | | | |
| Overall | 57.4 | 53.1 (44.1–62.8) | 60.5 (53.9–66.7) | 7.4 (-4.0 to 18.8) | 1.4 (0.8 −2.2) |
| 15–17 Years | 38.8 | 24.2 (8.0–45.9) | 44.7 (30.7–60.4) | 20.5 (-2.8 to 42.6) | 2.5 (0.9 −10.7) |
| 18–22 Years | 62.5 | 61.0 (50.5–71.0) | 64.8 (57.2–71.9) | 3.8 (-8.3 to 17.2) | 1.2 (0.7 −2.1) |
| **c) Sub group analysis: among those in school at baseline[c]** | | | | | |
| Overall | 97.2 | 98.3 (94.4–99.5) | 97.3 (95.9–98.7) | -1.0 (-3.5 to 3.3) | 0.6 (0.1 −2.4) |
| 15–17 Years | 97.5 | 99.0 (97.4–99.7) | 97.3 (95.5–98.8) | -1.7 (-3.9 to 0.3) | 0.4 (0.1 −1.1) |
| 18–22 Years | 96.5 | 96.3 (89.7–99.3) | 97.3 (93.5–99.2) | 0.9 (-5.3 to 8.8) | (0.2−7.5) |
| **d) Sub group analysis: among those who had not completed lower secondary school at baseline (post-hoc)[c,d]** | | | | | |
| Overall | 63.3 | 62.8 (56.0–81.0) | 63.5 (58.5–68.8) | 0.7 (-17.6 to 7.3) | 1.0 (0.4 −1.4) |
| 15–17 Years | 82.2 | 80.6 (71.5–89.1) | 82.7 (78.0–87.4) | 2.0 (-7.8 to 12.3) | 1.2 (0.6 −2.1) |
| 18–22 Years | 15.8 | 18.0 (7.8–55.7) | 15.2 (8.8–23.1) | -2.8 (-40.5 to 8.6) | 0.8 (0.1 −2.4) |

[a]CI—Confidence Interval

[b]Propensity score (PS) model adjusted for age, pregnancy and marital history (composite), study site, highest grade completed at baseline, orphanhood status, self-assessed poverty, wealth quantile, food insecurity, length of stay in the demographic surveillance area, education level of the household head, and schooling status at baseline

[c]PS model adjusted for fewer variables in the sub-group analyses (reduced sample): age, pregnancy and marital history (composite), study site, wealth quantile, food insecurity, education level of the household head

[d]PS model further adjusted for schooling status at baseline.

Our finding of high levels of school enrolment at endline in the youngest cohort (10–14 years) is similar to findings from other studies [5, 54]. In countries with universal free primary education policies like Kenya, it is common to have high levels of school enrolment up to a certain age [22]. However, some gaps remain for a small minority of these early adolescents. Among those out of school at endline (~3%), the most cited reasons for being out of school were pregnancy and lack of school fees. More concerted efforts to identify and support these adolescents with school fees, mitigating teenage pregnancies, provision of sexual reproductive health education and services, and reducing sexual violence which is unacceptably quite common in these study settings [55], are crucial.

The majority of DREAMS beneficiaries accessed multiple primary interventions, which may have all acted in different ways to influence schooling. The wider DREAMS interventions,

including those delivered in safe spaces, aimed to address the multiple sources of vulnerability among AGYW, in addition to more direct support through educational subsidies. Among AGYW in school at baseline, proportions accessing educational subsidies were significantly higher among DREAMS beneficiaries (56%) compared to non-beneficiaries (20%), indicating that DREAMS expanded access to educational subsidies and reached AGYW who would have otherwise attained less schooling and/or left school earlier. However, the funding available for education subsidies as part of DREAMS was constrained, as these subsidies were secondary interventions, and this may have limited DREAMS' impact. Given the many vulnerabilities in these two informal settlements, there is an opportunity to expand educational subsidies to reach more AGYW and to provide them with more support so as to enable higher levels of secondary school attainment.

About 40% of the AGYW who were aged 15–22 years and out of school at baseline had not achieved any post-primary training, indicating a bottleneck in transitioning from primary to secondary school or vocational training. While DREAMS enabled some re-enrolments, most AGYW did not re-enrol. Multiple reasons such as academic (un)readiness, school related costs, competing roles in the household, and even lack of role models [14, 33, 54, 56] which are common in these study settings [39, 55], all could impede any re-enrolment efforts. This suggests that further strategies to encourage, motivate and enable AGYW to stay or re-enrol back to school are warranted, for example through continued engagement with DREAMS mentors.

Among AGYW aged 15–22 years at baseline, overall, the proportion enrolled in school and/or completed lower secondary education at endline was high. While we only found weak evidence of an impact of DREAMS on educational attainment, the estimated increases were in a positive direction and were generally larger among AGYW aged 15–17 years (which more closely aligns with school-going age) compared with those 18–22 years at baseline. Greater effects among the younger age group, particularly among those out of school at baseline suggests that any DREAMS' impact was mainly through re-enrolments. Another possible explanation for the overall modest effect of DREAMS is that the vast majority (≥95%) of AGYW who were in school at baseline were still in school or had completed lower secondary school at endline in 2019, irrespective of whether or not they were a DREAMS beneficiary. DREAMS reached a high proportion of those in school at baseline (79%) (e.g., many of those who were food insecure), and it is possible that some DREAMS beneficiaries would have left school before completing lower secondary education in the absence of being a DREAMS beneficiary. In other words, there may have been differences at baseline between DREAMS and non-DREAMS beneficiaries that we did not measure. This may have resulted to some residual confounding in our causal analyses if these differences were important in determining schooling outcomes. In addition, as many AGYW were still in school at endline, longer follow up, is needed to know their 'final' outcomes and to understand the impact of DREAMS better, especially for the younger AGYW.

Only a few studies have evaluated the impact of multi-sectoral packages on educational outcomes. A four-arm randomized controlled trial incorporating violence, education, health and wealth creation interventions among girls aged 11–14 years found that the interventions increased rates of primary school completion and transition to secondary school by 5% when compared to the control arm in Kibera, two years after the intervention ended. Qualitative data from the same study indicates that the education component increased the girls' motivation for studying, and facilitated transfers to better-quality schools or boarding schools. However, some girls switched back to lower quality schools because they could not afford to pay the entire school fee after the program ended [21]. In western Kenya, a joint intervention that included an education subsidy (in form of school uniforms), and a school-based HIV prevention program had a smaller impact on primary school drop-out rates when compared to the

stand-alone education subsidy program [18]. These findings support the idea that educational subsidies can play an important role in enabling AGYW to transition to secondary school.

Our findings complement this small body of research evaluating multi-component interventions. Our study assesses a much larger intervention with multiple components, scaled up to reach many AGYW and implemented in a 'real-world context'. While our findings indicate only modest impact, they suggest that investments in education may be more effective in sustaining school enrolment and encouraging re-enrolments if introduced when AGYW are younger, during the transition period from primary to secondary school and before other obligations become more pressing. Older AGYW who already have competing priorities like caring for children or engaging in income-generating activities are likely to forgo re-joining formal education, and for them expanding other programs like vocational training and economic opportunities is warranted. DREAMS provided a comprehensive package to address multiple vulnerabilities among AGYW, and impacts on educational attainment may become more evident in the longer term, especially among those who were reached by DREAMS when they were relatively young. Continued long-term programming is needed to sustain the momentum for AGYW during the transitions from primary to secondary school.

## Strengths & limitations

Key strengths of this study include the study design, which leveraged existing health and demographic surveillance platforms, facilitating recruitment of large representative samples of AGYW for generalisability of the results. Results from the various methodological approaches in the sensitivity analyses were consistent with those from our primary analysis, indicating that the overall causal estimates were robust. While we relied on self-reported data, various strategies ensured that the risk of misclassifying some participants was minimised. First, DREAMS interventions in each setting were coordinated through a single implementing partner and through safe spaces, meaning that AGYW would know whether they had been invited to DREAMS. Second, we used objective measures of educational attainment. Third, the longitudinal nature of our data strengthened our measures, in that responses in each year were complemented by responses in subsequent years. Lastly, well-trained researchers collected the data, and this reduced the possibility of mis-reporting.

Limitations included differential loss to follow-up by AGYW characteristics, potentially contributing to selection bias. Although our cohort retention rates were high, and we controlled for confounding variables measured at enrolment in all our analyses, it is possible that outcomes were different among individuals who were followed up compared with those who were not. Our evaluation baseline took place in early 2017, months after DREAMS interventions had started. There is a possibility that some of the confounding variables that were measured at cohort enrolment might already have been impacted by DREAMS. However, the potential bias arising from this is likely minimal, because DREAMS interventions took time to roll-out, scale up and take effect [40]. It is unlikely that those who participated in the early stages of implementation (2016) had achieved a "sustained participation" sufficient to influence key confounding variables/outcomes by the time we collected our enrolment data.

We did not collect data on 'quality of schooling', often measured by test scores (e.g., reading or maths proficiency), as has been used by other studies [57]. The small sample size for the sub-group analysis, especially among those out of school at baseline, affected the precision of these estimates, and will have limited our ability to detect differences due to DREAMS. Nevertheless, the results are consistent with the full sample, and suggest that it may be worth investigating the impact of DREAMS in a larger sample of those who were out of school when they were first invited to DREAMS.

## Conclusions

The impact of DREAMS on educational attainment was modest, though estimated to be in a positive direction, and it was larger among younger AGYW. This age group (15–17 years) reflects the transition period between completing primary, and joining secondary school, and is an important window to potentially influence educational outcomes. While many AGYW received multiple interventions, there remains an opportunity to reach more of them with educational subsidies and thereby achieve a larger impact. Longer-term follow-up of the younger AGYW, many of whom were still in school at the last follow-up in 2019, would be valuable to better understand the ultimate educational attainment of DREAMS beneficiaries compared with non-beneficiaries.

## Supporting information

**S1 Fig. Kenya DREAMS layering table.**
(TIF)

**S2 Fig. Directed acyclic graphs to identify confounders for association between being a DREAMS beneficiary and educational attainment.**
(TIF)

**S3 Fig. Distribution of the estimated propensity scores among DREAMS beneficiaries and non-beneficiaries.**
(TIF)

**S4 Fig. Aspirations about schooling among AGYW aged 15–22 years.**
(TIF)

**S5 Fig. Educational attainment at baseline and endline* among AGYW out of school at baseline.** *Proportions at endline include those currently in school, as some participants re-enrolled during the follow-up.
(TIF)

**S1 Text. An extract of the interview questions among AGYW aged 15–22 years (English).**
(DOCX)

**S2 Text. Causal interpretation and sensitivity analyses.**
(DOCX)

**S1 Table. Number and proportions of AGYW aged 15–22 years retained in the study vs those lost to follow up at endline (2019), by AGYW characteristics.**
(XLSX)

**S2 Table. Number of interventions accessed by age group, schooling status at baseline, and invitation to DREAMS.**
(XLSX)

**S3 Table. Summary characteristics of AGYW by schooling status at baseline and by invitation to DREAMS.**
(XLSX)

**S4 Table. Results from sensitivity analysis for associations between DREAMS and educational attainment.**
(XLSX)

**S1 Dataset. Analytical datasets.**
(XLSX)

## Acknowledgments

We acknowledge the residents of Viwandani and Korogocho for their continued support and participation in the NUHDSS activities. We are grateful to the support provided by the project field team, the NUHDSS, and data management staff at APHRC. SM wishes to acknowledge International AIDS Vaccine Initiative (IAVI), and the University of California, San Francisco's International Traineeships in AIDS Prevention Studies (ITAPS), U.S. NIMH, R25MH123256 for writing support through a scientific writing workshop. We also acknowledge the role of Daniel Carter during 2018–2019, who worked as a study statistician and as a member of the study team's Data Analysis Working Group to co-develop methods of analysis and their programming in Stata.

## Author Contributions

**Conceptualization:** Sarah Mulwa, Abdhalah Ziraba, Sian Floyd.

**Data curation:** Sarah Mulwa.

**Formal analysis:** Sarah Mulwa.

**Funding acquisition:** Isolde Birdthistle, Sian Floyd.

**Methodology:** Sarah Mulwa, Isolde Birdthistle, Abdhalah Ziraba, Sian Floyd.

**Resources:** Lucy Chimoyi, Jane Osindo.

**Supervision:** Abdhalah Ziraba, Sian Floyd.

**Validation:** Jane Osindo.

**Writing – original draft:** Sarah Mulwa, Isolde Birdthistle, Sian Floyd.

**Writing – review & editing:** Sarah Mulwa, Lucy Chimoyi, Schadrac Agbla, Jane Osindo, Elvis O. Wambiya, Annabelle Gourlay, Isolde Birdthistle, Abdhalah Ziraba, Sian Floyd.

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
