## [Decision Letter · Decision Letter 0]

8 Apr 2021

PONE-D-21-08185

Impact of the DREAMS interventions on educational attainment among adolescent girls and young women: causal analysis of a prospective cohort in urban Kenya

PLOS ONE

Dear Dr. Mulwa,

Thank you for submitting your manuscript to PLOS ONE. After careful consideration, we feel that it has merit but does not fully meet PLOS ONE’s publication criteria as it currently stands. Therefore, we invite you to submit a revised version of the manuscript that addresses the points raised during the review process.

Based on the reviewers assessment the article needs considerable improvement in various aspects, particularly in clarity, regarding the design of the DREAMS interventions, enrolment and the various issues indicated. It seems the design did not include a randomization component and it is not clear, as a result, what the control group is.

In addition, it is reported that all the data is included in the manuscript, but this is not true. You are including the results of the analysis, not the database. With respect to this, you have two options:

- If the data are held or will be held in a public repository, include URLs, accession numbers or DOIs.

- If you include the data in the manuscript, state that "All relevant data are within the manuscript and its Supporting Information files.", but only when this is really the case.

We look forward to receiving your revised manuscript.

Kind regards,

José Antonio Ortega, Ph.D.

Academic Editor

PLOS ONE

Journal Requirements:

2. Please include additional information regarding the interview guide used in the study and ensure that you have provided sufficient details that others could replicate the analyses. For instance, if you developed an interview guide as part of this study and it is not under a copyright more restrictive than CC-BY, please include a copy, in both the original language and English, as Supporting Information

Reviewers' comments:

Reviewer's Responses to Questions

**Comments to the Author**

1. Is the manuscript technically sound, and do the data support the conclusions?

Reviewer #1: Yes

Reviewer #2: Yes

Reviewer #3: Partly

2. Has the statistical analysis been performed appropriately and rigorously? 

Reviewer #1: Yes

Reviewer #2: I Don't Know

Reviewer #3: No

3. Have the authors made all data underlying the findings in their manuscript fully available?

Reviewer #1: Yes

Reviewer #2: Yes

Reviewer #3: Yes

4. Is the manuscript presented in an intelligible fashion and written in standard English?

Reviewer #1: Yes

Reviewer #2: No

Reviewer #3: Yes

5. Review Comments to the Author

Reviewer #1: This study makes an important contribution to the literature by looking at the effect of participation in DREAMS on education among AGYW. These data are much anticipated and really add to the knowledge base on combination intervention strategies, which is much needed. This paper should be published but I have some comments on a few minor details.

Major

• It is noted that there was loss to follow up and that this was differential by participation in the DREAMS program. Is it possible to use weights or another strategy to determine how this might have affected your estimates?

• There are several mentions of the importance of educational subsides, yet this component of the package was not examined independently nor were other components. My impression is that this is not possible to untangle given the structure of the DREAMS data, but more discussion is needed on this topic. This is especially true given the weak effect on educational attainment. It would be important to know if girls who participated in more components or specific components had stronger effects or if we should be targeting specific subpopulations with larger programs like DREAMS.

• Causal assumptions were not mentioned including exchangeability, positivity, and consistency. In particular, I am worried that the assumption of consistency might not be met in this analysis as DREAMS beneficiaries may have included a range of different experiences with DREAMS including different types of interventions and frequency. If these are not met, then it may be safer to tone down causal language.

• The causal modeling approach that was used seems to be a hybrid of methods that involve propensity scores and g-computation. Can you provide a citation and more of an explanation of your rationale for this approach? More detail is also needed to explain this sentence: “These two logistic regression models were then used to predict the probability of attainment 3 for all AGYW, first under the scenario that they were not a DREAMS beneficiary, and second under the scenario that they were a DREAMS beneficiary.” Do you mean that you used the regression coefficients from the first model and set exposure to 1 and 0 in the model?

Minor

• Additional specific policy and programmatic recommendations are needed in the discussion section if possible. Several points such as the need for comprehensive and tailored interventions or educational subsidies seem like they were already incorporated within the structure of DREAMS or were not examined here.

• It was not mentioned if those who had already completed education at baseline were excluded, although I assumed that they were.

• Some confounders could be mediators (e.g. pregnancy). Are these all measured before DREAMS participation?

• I was unable to download the appendix so this may have been provided but it would be helpful to see a comparison of IPTW and your method in terms of results.

• The background mentions that most studies on cash transfers are randomized trials, but does this include literature about unconditional cash transfers or government grants?

Reviewer #2: Overall Comments:

1. Grammar and scientific writing style require improvements. For example, in the abstract, “impact on HIV incidence” should be “impact HIV incidence”. Likewise, “there was a suggestion” is inconsistent in terms of tense and is unclear. Are the authors making a suggestion? “Result to” should be “result in”, etc.

2. Are there other outcomes of interest besides educational attainment? Perhaps other psychosocial outcomes of interest, if the primary outcomes are reported elsewhere? Or other targets of the intervention that could be assessed, like educational goals, return to school, access to services/resources, or self-efficacy? Or gender equity? Including additional outcomes would significantly strengthen the manuscript.

3. In the methods section, it is unclear what the authors mean when the say “we descriptively summarized the educational status…”. Did the authors collect administrative data on school attendance, performance, re-enrollment, or was this entirely self-report? Reliance solely on self-report runs a high-risk of bias, particularly when this type of data should be collected by the schools.

4. How were the questions regarding aspirations and expectations developed? Is this a validated measure? If developed by the authors, were psychometrics assessed? If this construct was also explored in analyses, the authors should indicate this in the introduction as an additional outcome of interest besides educational attainment.

5. Were there any differences on demographics or other characteristics between participants who dropped out and those who completed DREAMS?

6. The results section is lengthy and difficult to follow. It would strengthen the manuscript to better organize the results, present them more concisely, and refrain from repeating findings in text that are presented in tables.

7. Was the sub-group analysis post-hoc? This should be clarified.

8. Confidence intervals are not reported in the correct format.

9. Overall, the manuscript provides modest evidence for the impact of DREAMs on girls educational attainment. It is unclear what implications the modest findings have beyond the current project in Kenya, or whether there are any particular recommendations or directions for future research based on the findings, besides the suggestion of longer-term follow-up data

Reviewer #3: My main concern is that the analysis does not generate "causal" estimates.

Here are the main concerns:

(1) The intervention was not randomized. Everyone was invited to participate. The Implementing Partners arguably invited the most vulnerable first, so there was a somewhat "phased-in" invitation process, but this is not well documented. Ultimately, those invited and those not invited in the first phase (by 2018) are very different at baseline.

(2) The authors attempt to adjust for selection into the treatment is through "propensity score matching". this is not fleshed out well. I would need to see the distribution of the propensity scores and how they overlap between the two groups. I could not see the results of the simple propensity score matching analysis. Table 5 was confusing.

(3) Most of the tables in the paper show summary statistics comparing the two groups (never invited vs. invited by 2018) but this comparison is flawed. Using so much space to document unadjusted differences when we know these unadjusted differences are not providing causal estimates does not seem the right call.

(4) There is differential attrition. This is shown in table S1. Those not invited are 11 percentage points more likely to be lost to follow-up (p<0.001). This is another threat to the validity of the analysis. Doing the propensity score matching among those found at endline is supposed to address this selection as well as the initial selection, but that analysis is not fletched out enough, as mentioned above. The attrition is substantial given the short time frame so I would not list "low attrition" as a strength of the study.

(5) The researchers do not have administrative data on who was invited or not. They have to rely on self-reports. This could be a poor proxy for actual invitation status. E.g. if those invited do not want to admit they were invited if they did not participate at all. The result that <99% of those who report they were invited participated in at least one activity is suggestive of this happening. It is extremely rare for take-up of a program to be >99%.

(6) The DREAMs intervention has many components. It is not very clear which ones were taken up by whom. A table showing take-up stats would be very helpful.

(7) The program started in March 2016 yet the baseline took place in March-July 2017.

(8) The text mentioned an acyclic graph but I could not find it.

6. PLOS authors have the option to publish the peer review history of their article (what does this mean?). If published, this will include your full peer review and any attached files.

Reviewer #1: No

Reviewer #2: No

Reviewer #3: No

---

## [Author Response · Author response to Decision Letter 0]

20 Jun 2021

Manuscript identifier: PONE-D-21-08185

Manuscript title: Impact of the DREAMS interventions on educational attainment among adolescent girls and young women: causal analysis of a prospective cohort in urban Kenya

Authors: Sarah Mulwa, Lucy Chimoyi, Schadrac Agbla, Jane Osindo, Elvis O. Wambiya, Annabelle Gourlay, Isolde Birdthistle, Abdhalah Ziraba, Sian Floyd

18th June 2021

Dear Dr. José Antonio Ortega,

Thank you for your response to our submission of the manuscript "Impact of the DREAMS interventions on educational attainment among adolescent girls and young women: causal analysis of a prospective cohort in urban Kenya" (PONE-D-21-08185). We appreciate the opportunity to address the comments raised by the reviewers. 

We have included a point-by-point response to accompany our manuscript, which has been further revised in accordance with the comments from the editor and the reviewers. We are pleased that the reviewers saw the value of this study, and recommended ways to enhance clarity. We have adopted these recommendations, clarified what our comparison groups are, and shortened in places while preserving key information in the text. We have included both a version with track changes as well as a clean version of the manuscript. Line numbers that we refer to in our response letter correspond to the revised clean version of the manuscript. 

Yours sincerely

Sarah Mulwa (on behalf of all co-authors) 

From the editor

Thank you for submitting your manuscript to PLOS ONE. After careful consideration, we feel that it has merit but does not fully meet PLOS ONE’s publication criteria as it currently stands. Therefore, we invite you to submit a revised version of the manuscript that addresses the points raised during the review process.

Based on the reviewer’s assessment the article needs considerable improvement in various aspects, particularly in clarity, regarding the design of the DREAMS interventions, enrolment and the various issues indicated. It seems the design did not include a randomization component and it is not clear, as a result, what the control group is.

In addition, it is reported that all the data is included in the manuscript, but this is not true. You are including the results of the analysis, not the database. With respect to this, you have two options:

- If the data are held or will be held in a public repository, include URLs, accession numbers or DOIs.

- If you include the data in the manuscript, state that "All relevant data are within the manuscript and its Supporting Information files.", but only when this is really the case.

***Thank you for the feedback. We have taken these comments into account and responses to each item are summarised below. Regarding availability of data, we have indicated that data underlying published results will be accessible and open at (https://microdataportal.aphrc.org/index.php/catalog), subject to a transition period as per the Open Access Policy of the Bill & Melinda Gates Foundation.

Journal Requirements:

2. Please include additional information regarding the interview guide used in the study and ensure that you have provided sufficient details that others could replicate the analyses. For instance, if you developed an interview guide as part of this study and it is not under a copyright more restrictive than CC-BY, please include a copy, in both the original language and English, as Supporting Information

**We have now clarified that the tools used in this study were developed by the research team (Methods, lines 185-186, page 8). We have included an extract of the questions used among 15-22 year olds as Supporting Information (S1 Text). The tool for 10-14 years has been published elsewhere (Mulwa et al., 2021)

**

Reviewer comments:

Reviewer #1: 

This study makes an important contribution to the literature by looking at the effect of participation in DREAMS on education among AGYW. These data are much anticipated and really add to the knowledge base on combination intervention strategies, which is much needed. This paper should be published but I have some comments on a few minor details.

Major

• It is noted that there was loss to follow up and that this was differential by participation in the DREAMS program. Is it possible to use weights or another strategy to determine how this might have affected your estimates?

*** There was indeed differential attrition by invitation to DREAMS at baseline, largely among AGYW aged 18-22 years (S1 Table). Due to various reasons (see below), we are unable to use any of the proposed approaches to determine how the attrition affected our estimates. However, we tried to minimise such biases in all our analyses by controlling for many of the demographic and socio-economic variables to account for any differences at baseline. In the Discussion, we have now included potential limitations related to this: While we accounted for many of the measured variables, we cannot rule out the possibility of unmeasured confounders (Discussion, lines 560-569, page 28). We also do not know much about those who we did not follow up, and outcomes may have been different among individuals who were followed up compared with those who were not (Strengths and limitations, lines 618-621, page 30).

Different approaches – including inverse probability weights, sensitivity bounds, and learning about selection through those it takes more effort to find – have been proposed to handle bias due to attrition. To use weights, we would assume that selective attrition is only related to observed variables. Since we do not have outcome information or updated explanatory variables for those who could not be reached, weighting by the baseline characteristics would not help us solve selection bias by much, or assess how this differential loss-to-follow-up affects our estimates. Sensitivity bounds on the other hand requires making assumptions about outcomes for those not followed up (for instance, assuming that all those who were not interviewed had the outcome of interest). An alternative approach is to use any available information about those not interviewed to understand a bit more about the selection. While we recorded the reasons for loss to follow-up at each interview round, the utility of this information is limited, although we do know that the majority of those not followed up were still residents (In 2019 (34/229) had emigrated while (168/229) could not be reached even after multiple attempts were made (but were still residents)*** 

• There are several mentions of the importance of educational subsides, yet this component of the package was not examined independently nor were other components. My impression is that this is not possible to untangle given the structure of the DREAMS data, but more discussion is needed on this topic. This is especially true given the weak effect on educational attainment. It would be important to know if girls who participated in more components or specific components had stronger effects or if we should be targeting specific subpopulations with larger programs like DREAMS.

*** We have now tried to be more clear in the Introduction (lines 120-122, page 6) that, since DREAMS was designed by PEPFAR as a complex intervention comprised of evidence-based components working synergistically, we sought a priori to evaluate the combined effect of the DREAMS ‘core package’ rather than individual effects of each component (for which there was already an evidence base). We still mention the reach and importance of educational subsidies because it is one of the components likely to have a direct influence on education. Among DREAMS beneficiaries, the proportions who received this specific component ranged from 20% among 18−22 year olds, to 57% among 10−14 years olds. Given that AGYW in our study settings face many economic challenges, we reflect on this issue in the Discussion (lines 534-542, page 27) by noting that it may help to reach a higher proportion of DREAMS beneficiaries with educational support.

In response to the issue of whether more components had stronger effects, we have now added the results from a sensitivity analysis based on the number of primary interventions received (exposure categories were: (i) never invited, (ii) invited and accessed <3 primary interventions, and (iii) invited and accessed ≥3 primary interventions) (S4 Table). There were little differences in the magnitude of impact of DREAMS when comparing groups (ii) and (iii) with group (i). This is not surprising, as the majority of those invited had accessed multiple interventions (e.g., >93% of 15-17 year olds accessed ≥2 primary interventions) (S2 Table) 

• Causal assumptions were not mentioned including exchangeability, positivity, and consistency. In particular, I am worried that the assumption of consistency might not be met in this analysis as DREAMS beneficiaries may have included a range of different experiences with DREAMS including different types of interventions and frequency. If these are not met, then it may be safer to tone down causal language.

*** We have now tried to describe the extent to which causal assumptions, including consistency, have been met in file S2 Text, which has been revised for clarity. The consistency assumption requires that exposure to DREAMS is clearly defined, such that any variations in receiving DREAMS would not result in a different outcome. Our analyses used a clear definition of exposure to DREAMS: invited yes or no. DREAMS implementation was based on a coherent core-package of interventions, and context specific adaptations were allowed. The impact of this heterogeneity on educational attainment is likely to be minimal given the methods of intervention delivery in Nairobi, where the delivery of DREAMS was fairly consistent in each setting. Implementation was coordinated by one implementing partner over the same time frame, and prioritization strategies to recruit the most vulnerable girls and young women evolved in a similar way across the two settings (Methods, lines 159-161; page 7; lines 169-172, page 8). With these strategies, we believe that the assumption of consistency holds in our analysis. *** 

• The causal modeling approach that was used seems to be a hybrid of methods that involve propensity scores and g-computation. Can you provide a citation and more of an explanation of your rationale for this approach? More detail is also needed to explain this sentence: “These two logistic regression models were then used to predict the probability of attainment 3 for all AGYW, first under the scenario that they were not a DREAMS beneficiary, and second under the scenario that they were a DREAMS beneficiary.” Do you mean that you used the regression coefficients from the first model and set exposure to 1 and 0 in the model? 

***Thank you for your feedback. We have now added the rationale for our approach, and included two references that guided us in thinking through our analytical approach (Lee & Little, 2017; Williamson, Morley, Lucas, & Carpenter, 2011) (Methods, line 271-274, page 12). In describing the causal inference approach, we have also modified our language to provide greater clarity and more detail (Methods, lines 271-289, page 12). 

Briefly, the two methods: propensity scores and g-computation are within the general framework of estimating causal effects from observational data, allowing us to compute the average treatment effects based on counterfactual scenarios using regression coefficients from the fitted models. With g-computation, outcome regression, i.e., a model for the outcome on treatment and all the observed covariates is run, then predictions under the counterfactual scenarios are computed. In our case, we used propensity score (PS) regression adjustment, i.e., a model for the outcome only on the PS and age (the PS was obtained by estimating fitted probabilities from a model of ‘invitation to DREAMS’ against all the relevant covariates. This approach is described in detail in the methods (Methods, 271-289, page 12). Our rationale for using propensity scores (PS) was its flexibility, and the fact that the approach is often robust to model misspecification compared to outcome regression models. In addition, the PS approach reduces the number of explanatory variables (and therefore the number of regression parameters) estimated from the model (our study is of modest sample size with a considerable number of confounding variable to adjust for (Methods, line 271-274, page 12). 

In describing the causal inference approach, we have now made it clear that two regression models were fit: (i) a logistic regression model of the outcome, age group and the propensity score; first among those who were invited to DREAMS (scenario 1) – from this model, we predicted the probability of attainment 3 (the outcome) for all AGYW, irrespective of whether or not they were invited to DREAMS. The average value of these probabilities was used to estimate the percentage of AGYW with attainment 3 under the counterfactual scenario that all AGYW were DREAMS beneficiaries, and (ii) a logistic regression model of the outcome, age group and the propensity score among those who were not invited to DREAMS (scenario 2) – in a similar manner, the average value of the probabilities from this model was used to estimate the percentage of AGYW with attainment 3 under the counterfactual scenario that all AGYW were not DREAMS beneficiaries (Methods, lines 280-289, page 12). *** 

Minor

• Additional specific policy and programmatic recommendations are needed in the discussion section if possible. Several points such as the need for comprehensive and tailored interventions or educational subsidies seem like they were already incorporated within the structure of DREAMS or were not examined here.

*** We have added additional recommendations in the Discussion. Some of these include: more concerted efforts among very young adolescents (10-14) to identify and support the small minority who were out of school (Discussion, lines 525-528, pages 26-27). Very few AGYW out of school at baseline went back to school even in the presence of DREAMS, indicating the need to identify barriers to re-enrolment, especially among older AGYW. Further strategies to encourage and motivate AGYW to stay or re-enrol back to school are warranted e.g., through continued engagement with DREAMS mentors (Discussion, lines 550-552, page 27). *** 

• It was not mentioned if those who had already completed education at baseline were excluded, although I assumed that they were.

*** We included everybody in the analysis. We set out to first understand levels of educational attainment in the study population through exploratory analysis, which then informed the ‘best’ realistic outcome definition that captures the reality in our study settings. DREAMS aimed to support those out of school to re-enrol back, while supporting those already enrolled in school remain in school. Based on the Kenya’s education system, on average, an individual aged 22 years (the upper age category in our study at baseline) would still be in institutions of higher education. For these reasons, we included everyone in the analysis irrespective of what level of education they had completed at baseline. We conducted further analyses stratifying by whether or not one was in school at baseline to account for any differential impacts of DREAMS by schooling status at baseline. 

Conducting an analysis excluding those who had completed lower secondary education at baseline would not allow us to capture the “full” impact of DREAMS, as DREAMS could have supported girls who had completed lower secondary education stay in school and complete the remaining two years of secondary school, or facilitated re-enrolments. Nonetheless, we have considered this comment and conducted post-hoc analysis among those who had not completed lower secondary education at baseline. Findings indicate no effect of DREAMS (Table 4d, Table 5d)***. 

• Some confounders could be mediators (e.g. pregnancy). Are these all measured before DREAMS participation?

*** Our evaluation baseline took place in early 2017, months after DREAMS interventions had started (DREAMS was not randomised). There is a possibility that some of our confounding variables that were measured at cohort enrolment may have already been impacted by DREAMS, and we include this as a potential limitation (Strengths and limitations, lines 621-628, page 30). However, the potential bias arising from this is likely to be minimal. Research conducted in the early stages of implementation indicates that it took time to roll-out and scale up interventions, especially those that required adapting to local context and sustained engagement e.g., social asset building (Chimbindi et al., 2018). For this reason, it is unlikely that anyone who participated in the early stages of implementation (2016) had achieved sustained participation to influence key confounding variables/outcomes by the time we collected our baseline data.***

• I was unable to download the appendix so this may have been provided but it would be helpful to see a comparison of IPTW and your method in terms of results.

***These comparisons are provided in S4 Table. The results from the IPTW were consistent with the other methods used in the analyses. ***

• The background mentions that most studies on cash transfers are randomized trials, but does this include literature about unconditional cash transfers or government grants?

***We have now modified our language to reflect that our background incorporates both unconditional and conditional cash transfers (Introduction, lines 83-87, page 5). From our review of published literature, we found that the majority of the available literature on cash transfers and education outcomes focuses mostly on conditional cash transfers (for instance (Baird, Ferreira, Özler, & Woolcock, 2013; Bastagli et al., 2016)). Fewer studies/reports have evaluated unconditional cash transfers (Kilburn, Handa, Angeles, Mvula, & Tsoka, 2017; Mostert & Vall Castello, 2020; The Kenya CT-OVC Evaluation Team, 2012). Two of these utilised cluster experiments. We have included these references in the manuscript ***

Reviewer #2: Overall Comments:

1. Grammar and scientific writing style require improvements. For example, in the abstract, “impact on HIV incidence” should be “impact HIV incidence”. Likewise, “there was a suggestion” is inconsistent in terms of tense and is unclear. Are the authors making a suggestion? “Result to” should be “result in”, etc.

*** Thank you for your comment. We have revised the manuscript and eliminated such errors as much as possible. ***

2. Are there other outcomes of interest besides educational attainment? Perhaps other psychosocial outcomes of interest, if the primary outcomes are reported elsewhere? Or other targets of the intervention that could be assessed, like educational goals, return to school, access to services/resources, or self-efficacy? Or gender equity? Including additional outcomes would significantly strengthen the manuscript.

*** We agree that exploring other outcomes is important. We have modified our language under Methods to distinguish between these different education outcomes in the paper by first listing the outcomes before describing each of them (Methods, lines 204-206 page 9). We chose to focus the paper on just education related measures because (i) education is a key outcome in its own merit; and (ii) we planned to conduct analyses by various groups: overall, by age group, and by schooling status at baseline. In the current paper, we conducted causal analysis of DREAMS on educational attainment only. We also explored other education related outcomes (these analyses were descriptive for various reasons). For example, we analysed aspirations and expectations about schooling (aspirations were already high so little scope for DREAMS to influence). We also looked at re-enrolments (5% of the participants re-enrolled) during the follow up period, but given the small numbers, further causal analysis was not possible. We reflect on some of the outcomes in the Discussion where possible (Discussion, lines 549-552, page 27; lines 560-571, page 28). Self-efficacy, social support and gender norms outcomes are being analysed across three evaluation sites – Nairobi (urban Kenya), Gem (rural Kenya), and uMkhanyakude in South Africa and were therefore not included in this paper. ***

3. In the methods section, it is unclear what the authors mean when the say “we descriptively summarized the educational status…”. Did the authors collect administrative data on school attendance, performance, re-enrollment, or was this entirely self-report? Reliance solely on self-report runs a high-risk of bias, particularly when this type of data should be collected by the schools.

***We used the statement to mean that causal analysis was not conducted for educational status (i.e., whether or not in school). We have deleted ‘descriptively’ from the text. We did not collect any administrative data. In Methods, lines 253-254, page 11, we indicate that causal analysis was only conducted for educational attainment. In each year of interview, we asked the participants if they were in school or not, and what grade they had completed. We then used the responses to classify participants into various categories such as continued enrolment, re-enrolment, dropouts, and so on (Table 3). We agree that biases may arise from self-reports. However, utilising data across the three surveys, using a fairly objective measure of attainment, and the fact that data were collected by well-trained researchers all reduced the possibility of mis-reporting thereby minimising bias. We have reflected on these issues under Strengths and limitations (lines 609-616, page 30)***

4. How were the questions regarding aspirations and expectations developed? Is this a validated measure? If developed by the authors, were psychometrics assessed? If this construct was also explored in analyses, the authors should indicate this in the introduction as an additional outcome of interest besides educational attainment.

*** We have clarified in the Methods that the research teams developed the questionnaires in this study. Some of the measures including those on aspirations and expectations were informed by existing instruments that have been used and validated in various settings for instance (Kabiru, Mojola, Beguy, & Okigbo, 2013) (Methods, lines 185-186, page 8; lines 189-190, page 8-9). Still, we have taken this point into account and assessed the scale reliability for these items using our data (Methods, lines 226-227, page 10). As we did not use causal analysis for aspirations and expectations because of the reason summarised in point #2 (we only summarised these using descriptive statistics), we have not modified the language in the introduction. ***

5. Were there any differences on demographics or other characteristics between participants who dropped out and those who completed DREAMS?

***As this was an independent evaluation of DREAMS, we do not have program data on completion of DREAMS interventions. From our data however, we know that the majority of those invited to DREAMS accessed multiple primary interventions (95% accessed ≥2 out of 7; 85% accessed ≥3) – a proxy for sustained participation (S2 Table). Our analyses indicate some differences in characteristics at enrolment between DREAMS beneficiaries and non-beneficiaries, mainly by schooling status, age, and pregnancy history (Table 2). ***

6. The results section is lengthy and difficult to follow. It would strengthen the manuscript to better organize the results, present them more concisely, and refrain from repeating findings in text that are presented in tables.

***We have considered these suggestions and shortened the results section, presenting the findings more concisely (for instance we have reduced the text based on Fig 2)***.

7. Was the sub-group analysis post-hoc? This should be clarified.

*** We have now clarified in the Methods that the sub-group analysis by schooling status at baseline was pre-specified (Methods, line 296, page 13). Following a comment raised by reviewer #1, we have included a post-hoc sub-group analysis among those who had not attained lower secondary educational at baseline (Methods, line 299, page 13). 

8. Confidence intervals are not reported in the correct format.

*** We have now corrected the confidence intervals within the manuscript and the tables *** 

9. Overall, the manuscript provides modest evidence for the impact of DREAMs on girls educational attainment. It is unclear what implications the modest findings have beyond the current project in Kenya, or whether there are any particular recommendations or directions for future research based on the findings, besides the suggestion of longer-term follow-up data

*** We have added additional recommendations in the Discussion, applicable to the current project, as well as to contexts where resources are limited. Some of these include: more concerted efforts among very young adolescents (10-14) to identify and support the small minority who were out of school (Discussion, lines 525-528, pages 26-27). Very few AGYW out of school at baseline went back to school even in the presence of DREAMS, indicating the need to identify barriers to re-enrolment, especially among older AGYW. Further strategies to encourage and motivate AGYW to stay or re-enrol back to school are warranted e.g., through continued engagement with DREAMS mentors (Discussion, lines 550-552, page 27). *** 

Reviewer #3: 

My main concern is that the analysis does not generate "causal" estimates.

Here are the main concerns:

(1) The intervention was not randomized. Everyone was invited to participate. The Implementing Partners arguably invited the most vulnerable first, so there was a somewhat "phased-in" invitation process, but this is not well documented. Ultimately, those invited and those not invited in the first phase (by 2018) are very different at baseline

***We have now clarified in the Methods that not everyone was invited to participate in DREAMS. The implementing partners targeted and extended invitation to the most vulnerable individuals (e.g., those who were food insecure, out of school etc.) (Methods, lines 169-172, page 8). Adolescents and young women coming from these settings experience very many vulnerabilities, but only a subset of those meeting the targeting criteria were recruited given resource constraints. Invitation to participate in DREAMS continued into 2018 (restricted to those who met the vulnerability criteria), and intervention delivery continued during 2019-20 So, while the invitation was “phased-in”, not everyone was ‘targeted’ and eventually invited. 

Comparing those invited to those not invited by socio-demographic characteristics, differences were mainly observed for food insecurity (those reporting food insecurity were more likely to be invited) and age (older, ever married and ever pregnant AGYW less likely to be invited) (Table 2). As invitation was offered to some individuals and not others as described above, we used self-reported invitation to participate in DREAMS to classify participants as exposed or not in the absence of randomization. We have included the propensity score graphs (S3 Fig) which shows good overlap between those invited vs not invited, indicating that fair comparison between the two groups (beneficiaries vs non-beneficiaries) is possible; so long as we are adjusting for baseline characteristics). 

 ***

(2) The authors attempt to adjust for selection into the treatment is through "propensity score matching". this is not fleshed out well. I would need to see the distribution of the propensity scores and how they overlap between the two groups. I could not see the results of the simple propensity score matching analysis. Table 5 was confusing.

***We used propensity scores to adjust for imbalances between those who were invited vs not invited to DREAMS. We have now modified our language to include the rationale for our approach, and summarise how the propensity score analysis was implemented (Methods, lines 271-288, pages 12). The primary analysis approach was ‘propensity score regression adjustment’ and so are the results presented in Table 5. We did not conduct ‘propensity score matching’ as our interest was to evaluate DREAMS’ impact on educational attainment at the population-level, and not in the group who were actually invited (Williamson et al., 2011).

We have now included a supplementary figure with the distribution of the propensity scores between the two exposure groups. The figures show a good overlap of the scores between the two comparison groups (S3 Fig). We also found good covariate balance in the sensitivity analyses with inverse-probability-of-treatment weighting (with probability of treatment equal to the propensity score) (S2 Text). All these checks ensure that our analyses are robust to key assumptions. ***

(3) Most of the tables in the paper show summary statistics comparing the two groups (never invited vs. invited by 2018) but this comparison is flawed. Using so much space to document unadjusted differences when we know these unadjusted differences are not providing causal estimates does not seem the right call.

*** We have considered this suggestion, and where possible, we have shortened the descriptive text (e.g., summary of Fig 2). We agree that the summary statistics do not take into account confounding and do not provide causal estimates. However, we do believe that providing these summaries is important to better understand the observed levels of key outcomes and explanatory variables in the study population, as well as help us in interpretation of the key findings.***

(4) There is differential attrition. This is shown in table S1. Those not invited are 11 percentage points more likely to be lost to follow-up (p<0.001). This is another threat to the validity of the analysis. Doing the propensity score matching among those found at endline is supposed to address this selection as well as the initial selection, but that analysis is not fletched out enough, as mentioned above. The attrition is substantial given the short time frame so I would not list "low attrition" as a strength of the study.

*** We have taken into account your previous comments and described the propensity score analysis more clearly (Methods, lines 271-288, pages 12). To better understand the attrition rates, we have included the loss to follow-up separately for each age group, and the differences by invitation at baseline are largely driven by the older AGYW, who were often harder to reach and engage with DREAMS (S1 Table). We have excluded high retention from the strengths of the study. Although we controlled for confounding variables measured at enrolment in all our analyses, it is possible that outcomes were different among individuals who were followed up compared with those who were not, and we include the differential attrition as a potential limitation in our study (Strengths and imitations. Lines 617-620, page 30). *** 

(5) The researchers do not have administrative data on who was invited or not. They have to rely on self-reports. This could be a poor proxy for actual invitation status. E.g. if those invited do not want to admit they were invited if they did not participate at all. The result that <99% of those who report they were invited participated in at least one activity is suggestive of this happening. It is extremely rare for take-up of a program to be >99%.

*** We agree that self-reports may result to misclassifications. We acknowledge the concern that some people might have said they were not invited, just because they did not have access any intervention. However, we do not think self-reporting influenced our exposure definition very much. First, we found consistent data reporting in relation to invitation to DREAMS, with many of those who said they had been invited at baseline also saying they had been invited in 2018. Second, the questionnaire had skip patterns, with questions about awareness and invitation to DREAMS asked first, followed by questions about the specific interventions accessed. Lastly, DREAMS interventions were coordinated by one partner in each setting and through safe spaces, and this meant that AGYW would know if they had been invited or not. We have now reflected on these points in the Strengths and limitations section (lines 609-616, page 30). As a key principle of DREAMS was layering i.e., offering multiple interventions to those invited (with dedicated efforts to engage those who had been invited), it is not surprising that 99% of those invited reported accessing at least one primary intervention (out of 7) by 2019.

(6) The DREAMs intervention has many components. It is not very clear which ones were taken up by whom. A table showing take-up stats would be very helpful.

***The DREAMS interventions were conceptualised as a core-package, and the goal of this evaluation was to assess the effect of receiving the combined package. We do not think that presenting each of the components by socio-demographic characteristics within this paper is useful. The current summaries show that many of those invited accessed at least three primary interventions (S2 Table). Given we know (and show in the paper - Table 2) who was invited according to their characteristics, we have good insights into who was more likely to access the interventions (strong association between invitation and interventions), which we believe is sufficient for this purpose. More information on uptake of DREAMS, including uptake of each core package category, is documented in detail in another paper (currently under review; poster (http://programme.aids2020.org/Abstract/Abstract/7340) *** 

(7) The program started in March 2016 yet the baseline took place in March-July 2017.

***As indicated in the response to reviewer 1, all confounding variables included in the analyses, as well as participation in DREAMS were measured in 2017 in this evaluation. While we captured DREAMS participation in 2017, DREAMS implementation had started in 2016, and research in the early stages of implementation indicates that this took time to roll-out and scale up, especially for interventions that required adapting to local context and sustained engagement (Chimbindi et al., 2018). As it would have taken time for DREAMS to influence outcomes, it is unlikely that anyone who participated in the early stages of implementation (2016) had achieved a “sustained participation” to influence key confounding variables/outcomes by the time we collected our data. We still cannot rule out the possibility of some effect in the early stages of implementation, and we do reflect on this in the limitations (Strengths and limitations, lines 621-628, page 30). ***

(8) The text mentioned an acyclic graph but I could not find it.

***We have now included the DAG as one of the supplementary files (S2 Fig)***

End

---

## [Decision Letter · Decision Letter 1]

12 Jul 2021

Impact of the DREAMS interventions on educational attainment among adolescent girls and young women: causal analysis of a prospective cohort in urban Kenya

PONE-D-21-08185R1

Dear Dr. Mulwa,

We’re pleased to inform you that your manuscript has been judged scientifically suitable for publication and will be formally accepted for publication once it meets all outstanding technical requirements.

Kind regards,

José Antonio Ortega, Ph.D.

Academic Editor

PLOS ONE

Additional Editor Comments (optional):

Both reviewers and the academic editor feel that the suggested changes have been satisfactorily dealt with. Reviewer 2 from the first revision was unavailable, but the academic editor feels that their suggested changes were also addressed. Congratulations!

Reviewers' comments:

Reviewer's Responses to Questions

**Comments to the Author**

1. If the authors have adequately addressed your comments raised in a previous round of review and you feel that this manuscript is now acceptable for publication, you may indicate that here to bypass the “Comments to the Author” section, enter your conflict of interest statement in the “Confidential to Editor” section, and submit your "Accept" recommendation.

Reviewer #1: All comments have been addressed

Reviewer #3: All comments have been addressed

2. Is the manuscript technically sound, and do the data support the conclusions?

Reviewer #1: Yes

Reviewer #3: Yes

3. Has the statistical analysis been performed appropriately and rigorously? 

Reviewer #1: Yes

Reviewer #3: Yes

4. Have the authors made all data underlying the findings in their manuscript fully available?

Reviewer #1: No

Reviewer #3: No

5. Is the manuscript presented in an intelligible fashion and written in standard English?

Reviewer #1: Yes

Reviewer #3: Yes

6. Review Comments to the Author

Reviewer #1: (No Response)

Reviewer #3: (No Response)

7. PLOS authors have the option to publish the peer review history of their article (what does this mean?). If published, this will include your full peer review and any attached files.

Reviewer #1: No

Reviewer #3: No

---

## [Editor Report · Acceptance letter]

4 Aug 2021

PONE-D-21-08185R1 

Impact of the DREAMS interventions on educational attainment among adolescent girls and young women: causal analysis of a prospective cohort in urban Kenya 

Dear Dr. Mulwa:

I'm pleased to inform you that your manuscript has been deemed suitable for publication in PLOS ONE. Congratulations! Your manuscript is now with our production department. 

Kind regards, 

on behalf of

Dr. José Antonio Ortega 

Academic Editor

PLOS ONE